# Wasserstein Auto-encoders:
# latent dimensionality and random encoders

**Paul Rubenstein,**[*] **Bernhard Schölkopf, Ilya Tolstikhin**
Empirical Inference
Max Planck Institute for Intelligent Systems, Tübingen
{paul.rubenstein,bs,ilya}@tuebingen.mpg.de

## Abstract

We study the role of latent space dimensionality in Wasserstein auto-encoders (WAEs). Through experimentation on synthetic and real datasets, we argue that random encoders should be preferred over deterministic encoders.

## 1 Introduction

Wasserstein auto-encoders (WAEs) are a recently introduced auto-encoder architecture with justification stemming from the theory of Optimal Transport (Tolstikhin et al., 2018). Similarly to Variational auto-encoders (VAEs), WAEs describe a particular way to train probabilistic *latent variable models* (LVMs) $P_G$. LVMs act by first sampling a code (feature) vector $Z$ from a *prior distribution* $P_Z$ defined over the latent space $\mathcal{Z}$ and then mapping it to a random input point $X \in \mathcal{X}$ using a conditional distribution $P_G(X|Z)$ also known as *the decoder*.

Instead of minimizing the KL divergence between the LVM $P_G$ and the unknown data distribution $P_X$ as done by VAEs, WAEs aim at minimizing any optimal transport distance between them. Given any non-negative cost function $c(x, x')$ between two images, WAEs minimize the following objective with respect to parameters of the decoder $P_G(X|Z)$:

$$\min_{Q(Z|X)} \mathbb{E}_{P_X} \mathbb{E}_{Q(Z|X)} \big[ c\big(X, G(Z)\big)\big] + \lambda \mathcal{D}_Z(Q_Z, P_Z), \tag{1}$$

where the conditional distributions $Q(Z|X)$ are commonly known as *encoders*, $Q_Z(Z) := \int Q(Z|X)P_X(X)dX$ is *the aggregated posterior* distribution, $\mathcal{D}_Z$ is any divergence measure between two distributions over $\mathcal{Z}$, and $\lambda > 0$ is a regularization coefficient. In practice $Q(Z|X = x)$ and $G(z)$ are often parametrized with deep nets, in which case back propagation can be used with stochastic gradient descent techniques to optimize the objective. We will consider only *random* encoders $Q(Z|X = x)$ mapping inputs to a *distribution* over the latent space.

The objective (1) is similar to that of the VAE and has two terms. The first *reconstruction term* aligns the encoder-decoder pair so that the encoded images can be accurately reconstructed by the decoder as measured by the cost function $c$ (e.g. $l_2$ or *cross-entropy loss)*. The second regularization term is different from VAEs: it forces the aggregated posterior $Q_Z$ to match the prior distribution $P_Z$ rather than asking point-wise posteriors $Q(Z|X = x)$ to match $P_Z$ simultaneously for all data points $x$. This means that WAEs explicitly control the shape of the *entire* encoded dataset while VAEs constrain every input point separately.

In this work, we address one of the important design choices of WAEs related to the properties of the latent space which were not discussed in Tolstikhin et al. (2018) — whether to use deterministic encoders or random (probabilistic) encoders.

We illustrate different ways in which a mismatch between the latent space dimensionality $d_{\mathcal{Z}}$ and the intrinsic data dimensionality $d_{\mathcal{I}}$ may hurt the performance of WAEs and argue that WAEs can be made adaptive to the unknown intrinsic data dimensionality $d_{\mathcal{I}}$ by using random encoders rather than the deterministic encoders used in all experiments of Tolstikhin et al. (2018). The performance of random encoders is on par with the deterministic ones when $d_{\mathcal{Z}} \leq d_{\mathcal{I}}$, and potentially better when $d_{\mathcal{Z}} \gg d_{\mathcal{I}}$ which is typical for real-world applications, suggesting that random encoders should generally be preferred when using WAEs.

---

[*]Also affiliated with: Machine Learning Group, Engineering Department, University of Cambridge

Deterministic                                    Random

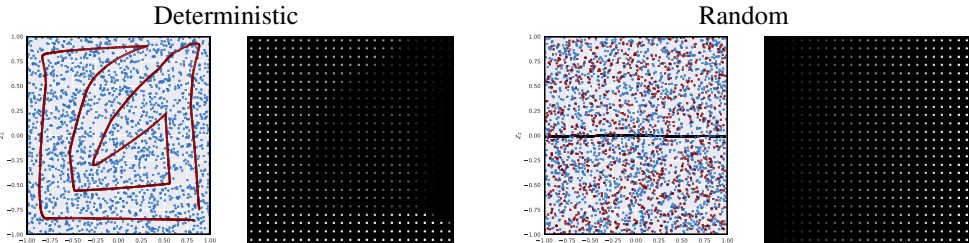

Figure 1: Illustrations of the 2-dimensional latent space of the WAE trained on the dataset with $d_{\mathcal{I}} = 1$ with deterministic (**left pair**) and random (**right pair**) encoders and a uniform prior $P_Z$ over the box; Within each pair: (**left**) 1000 points sampled from the aggregated posterior $Q_Z$ (**dark red**) and prior $P_Z$ (**blue**). For the random encoder **black** points show the mean values of the encoder $\mathbb{E}[Q(Z|X = x)]$; (**right**) decoder outputs at the corresponding points of the latent space.

## 2 DIMENSION MISMATCH IS HARMFUL FOR DETERMINISTIC ENCODERS

What happens if a deterministic-encoder WAE is trained with a latent space of dimension $d_{\mathcal{Z}}$ that is larger than the intrinsic dimensionality $d_{\mathcal{I}}$? We empirically investigated this setting by training WAEs on a simple synthetic dataset consisting of fixed-size and -position grey squares on a black background. Each image is identifiable from the colour of its square, which is uniformly distributed on the interval $[0, 1]$ and hence the intrinsic dimensionality of this dataset is 1.

We trained both random-[1] and deterministic-encoder WAEs with a latent dimension of 2. As such, it is possible to fully visualise both the learned encoder and decoder (see Figure 1). The deterministic-encoder WAE is forced to reconstruct the images well, while at the same time trying to fill the latent space uniformly with the 1-dimensional data manifold. This is not possible, but the WAE does the best it can by curling the manifold up in the latent space. The random-encoder WAE in contrast can use 1 latent dimension to encode useful information to the decoder, while filling the unneeded dimension with noise, thus succeeding in matching $Q(Z)$ and $P(Z)$.

To what extent is it actually a problem that the deterministic WAE represents the data as a curved manifold in the latent space? There are two issues.

**Poor sample quality**: Only a small fraction of the total volume of the latent space is covered by the deterministic encoder. Hence the decoder is only trained on this small fraction, because under the objective (1) the decoder learns to act only on the encoded training images. While it appears in this 2-dimensional toy example that the quality of decoder samples is nonetheless good everywhere, in high dimensions, such "holes" may be significantly more problematic. Indeed, Figure 2 (left) show that WAEs with deterministic encoders produce bad samples when $d_{\mathcal{Z}} \gg d_{\mathcal{I}}$

**Wrong proportion of generated images**: We found that although in this simple example all of the samples generated by the deterministic-encoder WAE are of good quality, they are not produced in the correct proportions. By analogy, this would correspond to a model trained on MNIST producing too few 3s and too many 7s.

## 3 RANDOM ENCODERS WITH LARGE $d_{\mathcal{Z}}$

To test our new intuitions about the behaviour of deterministic- and random-encoder WAEs with different latent dimensions, we next consider the *CelebA* dataset. All experiments reported in this section used Gaussian priors and, for the random-encoder WAEs, Gaussian encoders. A fixed convolutional architecture with cross-entropy reconstruction loss was used for all experiments. To keep computation time feasible, we used small networks.

Figure 2 (left) shows the results of training 5 random- and 5 deterministic-encoder WAEs with $d_{\mathcal{Z}} = 32, 64, 128$ and $256$. We found that both deterministic- and random-encoder WAEs exhibit very similar behaviour: the FID scores Heusel et al. (2017) of random samples generated by the models after training first decrease to some minimum and then subsequently increase (lower FID scores mean better sample quality).

---

[1] The random-encoder maps an input image to a uniform distribution over an axis aligned box in $\mathcal{Z}$.

| $d_{\mathcal{Z}}$ | FID score | |
|---|---|---|
| | Det. | Rand. |
| 32 | $75.0 \pm 0.7$ | $74.8 \pm 0.5$ |
| 64 | $71.6 \pm 0.8$ | $71.1 \pm 1.0$ |
| 128 | $76.8 \pm 1.3$ | $76.8 \pm 1.2$ |
| 256 | $147.6 \pm 2.3$ | $139.8 \pm 4.2$ |

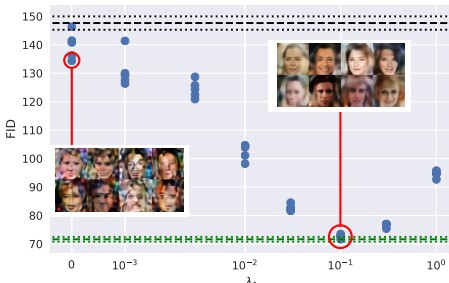

Figure 2: **Left:** FID scores for deterministic- and random-encoder WAEs trained on *CelebA* for various latent dimensions $d_{\mathcal{Z}}$. FID scores suffer for $d_{\mathcal{Z}} \gg d_{\mathcal{I}}$. **Right:** FID scores for random-encoder WAEs with latent space dimension $d_{\mathcal{Z}} = 256$ for different $L_1$ regularisation coefficients $\lambda_1$. Dashed/dotted black lines represent the mean $\pm$ s.d. for deterministic-encoder WAEs with $d_{\mathcal{Z}} = 256$. Dashed/dotted green lines represent the mean $\pm$ s.d. for deterministic WAEs $d_{\mathcal{Z}} = 64$, for which the FID scores were best amongst all latent dimensions we tested. Overlaid images are random samples coming from experiments indicated by the red circle. These plots show that when appropriately regularised ($\lambda = 10^{-1}$), random encoders with high dimensional latent spaces are able to produce samples of similar quality to the deterministic encoders with the best latent space dimension.

For deterministic encoders, this agrees with the intuition we gained from the previous experiment. Unable to fill the whole latent space when $d_{\mathcal{I}} < d_{\mathcal{Z}}$, the encoder leaves large holes in the latent space on which the decoder is never trained. When $d_{\mathcal{I}} \ll d_{\mathcal{Z}}$, these holes occupy most of the total volume, and thus most of the samples produced by the decoder from draws of the prior are poor.

For random encoders we did not expect this behaviour. Rather than automatically filling unnecessary dimensions with noise when $d_{\mathcal{I}} \ll d_{\mathcal{Z}}$ similarly to the previous experiment, thus making $Q_Z$ accurately match $P_Z$ and preserving good sample quality, the random encoders would "collapse" to deterministic encoders. That is, the variances of $Q(Z|X = x)$ tend to 0 for almost all dimensions and inputs $x$.

**Resolving variance collapse through regularization** The cause of this variance collapse is uncertain to us. We found that we could effectively eliminate it by adding additional regularisation in the form of an $L_1$ penalty on the log-variances, providing encouragement for the variances to remain closer to 1 and thus for the encoder to be remain stochastic. More precisely, we added the following term to the objective function to be minimised:

$$\frac{\lambda_1}{N} \sum_{n=1}^{N} \sum_{i=1}^{d_{\mathcal{Z}}} \left| \log \left( \sigma_i^2(x_n) \right) \right| \tag{2}$$

where $i$ indexes the dimensions of the latent space $\mathcal{Z}$, $n$ indexes the inputs in a mini-batch and $\lambda_1 \geq 0$ is a regularization hyper-parameter.

Using latent dimensions 256 to consider over-shooting the intrinsic dimensionality of the dataset we trained 5 $L_1$-regularized random-encoder WAEs for a variety of values for $\lambda_1$. Figure 2 shows the test reconstruction errors and FID scores obtained at the end of training. These results show that $L_1$ regularisation can significantly improve the performance of random-encoder WAEs compared to their deterministic counterparts. In particular, tuning for the best $\lambda_1$ parameter results in samples of quality comparable to deterministic encoders with the best latent dimension size. Through appropriate regularisation, random-encoder WAEs are able to adapt to the case that $d_{\mathcal{Z}} \gg d_{\mathcal{I}}$ and perform well.

## 4 CONCLUSION

The reader will notice that we have merely substituted the problem of searching for the "right" latent dimensionality $d_{\mathcal{Z}}$ with the problem of searching for the "right" regularisation $\lambda_1$. However, these results show that random encoders are capable of adapting to the intrinsic data dimensionality; future directions of research include exploring divergence measures other than MMD and whether the $L_1$ regularisation coefficient $\lambda_1$ can be adaptively adjusted by the learning machine itself.

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
