# OpenReview forum: "Wasserstein Auto-Encoders: Latent Dimensionality and Random Encoders"
_ICLR.cc/2018/Workshop — Accept_

### Official Review · AnonReviewer2 · 2018-02-22
**Accept**

**Rating:** 7
**Confidence:** 4

**Review:**

This paper is pointing a problem worth addressing in Wasserstein Autoencoders, how to exploit a medium to high dimensional latent space.
Section 3: the L1 penalty on the log variance does not just enforce stochasticity of the encoder but also encourages it to be closer to the prior in terms of variance.
The problem of using higher dimensional latent space effectively in WAE (where the performance seems to degrade) is very similar in Variational Autoencoders [1, 2] where the model behavior reverts into one of a smaller latent dimensionality model (as pointed in [3]) for similar reasons: rebuilding and filling the volume of a high dimensional standard gaussian with smaller gaussians is hard. Proposed solutions included using normalizing flows for stochastic variational inference [4] with more complex approximate posterior distributions for the encoder.
In that regard, the use of gaussian encoder in this paper seems like a missed opportunity: since there is no constraint of tractability (unlike stochastic variational inference [5]) using a more general form F(x, \epsilon) (e.g. with F a neural network)  for the encoder would still enable the computation of the cost function. Which brings up the question of whether this approach would solve this issue.

[1] Danilo Rezende, Shakir Mohamed, Daan Wierstra: Stochastic Backpropagation and Approximate Inference in Deep Generative Models. ICML 2014.
[2] Diederik Kingma, and Max Welling: Auto-Encoding Variational Bayes. ICLR 2014.
[3] Yuri Burda, Roger Grosse, and Ruslan Salakhutdinov: Importance Weighted Autoencoders. ICLR 2015.
[4] Danilo Jimenez Rezende, Shakir Mohamed: Variational Inference with Normalizing Flows. ICML 2015.
[5] John Paisley, David Blei, and Michael Jordan: Variational bayesian inference with stochastic search. ICML 2012.

---

> ### Public Comment · ~Paul_K_Rubenstein1 · 2018-03-21
> **Reply**
>
> Many thanks for your review and pointing us towards this relevant literature. Considering implicit noise distributions is indeed something of interest to us and we are currently working on this.

---

### Official Review · AnonReviewer1 · 2018-03-10
**Concise empirical study on deterministic vs stochastic encoders**

**Rating:** 7
**Confidence:** 4

**Review:**

The paper describes a well conducted and concise empirical study on choosing deterministic vs. stochastic encoders when training a Wasserstein Auto-Encoders. The experiments are small scale, systematic and sufficient to gain a better understanding of the tradeoffs involved. The results seem generally not very surprising, but I think it is important to publish them and make them accessible to the research community for future reference.

The scope of of this submission leaves a bit the impression that it could have been an appendix on the paper that originally introduced WAEs; but I think it also fits well into the 3 pages workshop-track format.

---

> ### Public Comment · ~Paul_K_Rubenstein1 · 2018-03-21
> **Reply**
>
> We were indeed unsurprised by the results showing that using deterministic encoders when the intrinsic data dimensionality is smaller than the latent dimensionality results in the latent space being improperly filled. However, we should point out that we found it somewhat surprising that in high dimensions the random encoders also fall prey to this ('variance collapse'). We provided one possible solution to this problem in the form of the regularisation we introduced, but whether there are better solutions is still an open question.

---

### Decision · Program_Chairs · 2018-03-20
**ICLR 2018 Workshop Acceptance Decision**

**Decision:**

Accept

**Comment:**

Congratulations, your paper was accepted to the ICLR workshop.